# American crocodiles (*Crocodylus acutus*) as restoration bioindicators in the Florida Everglades

Venetia S. Briggs-Gonzalez[1]⊕*, Mathieu Basille[1]⊕, Michael S. Cherkiss[2]⊕, Frank J. Mazzotti[1]⊕

**1** Department of Wildlife Ecology and Conservation, Fort Lauderdale Research and Education Center, University of Florida, Fort Lauderdale, Florida, United States of America, **2** U.S. Geological Survey, Wetland and Aquatic Research Center, Fort Lauderdale, Florida, United States of America

⊕ These authors contributed equally to this work.
* vsbriggs@ufl.edu

**Data Availability Statement:** Data available at Dryad Digital Repository Briggs-Gonzalez, Venetia (2020), American crocodile captures in South

## Abstract

The federally threatened American crocodile (*Crocodylus acutus*) is a flagship species and ecological indicator of hydrologic restoration in the Florida Everglades. We conducted a long-term capture-recapture study on the South Florida population of American crocodiles from 1978 to 2015 to evaluate the effects of restoration efforts to more historic hydrologic conditions. The study produced 10,040 crocodile capture events of 9,865 individuals and more than 90% of captures were of hatchlings. Body condition and growth rates of crocodiles were highly age-structured with younger crocodiles presenting with the poorest body condition and highest growth rates. Mean crocodile body condition in this study was 2.14 ±0.35 SD across the South Florida population. Crocodiles exposed to hypersaline conditions (> 40 psu) during the dry season maintained lower body condition scores and reduced growth rate by 13% after one year, by 24% after five years, and by 29% after ten years. Estimated hatchling survival for the South Florida population was 25% increasing with ontogeny and reaching near 90% survival at year six. Hatchling survival was 34% in NE Florida Bay relative to a 69% hatchling survival at Crocodile Lake National Wildlife Refuge and 53% in Flamingo area of Everglades National Park. Hypersaline conditions negatively affected survival, growth and body condition and was most pronounced in NE Florida Bay, where the hydrologic conditions have been most disturbed. The American crocodile, a long-lived animal, with relatively slow growth rate provides an excellent model system to measure the effects of altered hydropatterns in the Everglades landscape. These results illustrate the need for continued long-term monitoring to assess system-wide restoration outcomes and inform resource managers.

## Introduction

Evaluating the success of restoration efforts depends on effective monitoring programs and the use of ecological indicators that are representative of the system [1]. Indicators should demonstrate clear responses to system-wide changes and be effectively and efficiently

Florida, Dryad, Dataset, https://doi.org/10.5061/dryad.nzs7h44q7.

**Funding:** Funding for this work was supported by grants to FJM by U.S. Army Corps of Engineers (USACE https://www.usace.army.mil/) Cooperative Agreement Research Work Order #268, U.S. National Park Service (https://www.nps.gov/) CESU Cooperative Agreement #H5000060106, U. S. Geological Survey (USGS https://www.usgs.gov/) Greater Everglades Priority Ecosystems Science (PES) Program Cooperative Agreement #G15AC00278, Florida Power and Light Company (https://www.fpl.com/) Contract #02377545, Lacoste/Save Your Logo Fonds de Dotation pour la Biodiversité (http://www.saveyourlogo.org/en/) #F017137, U.S. Fish and Wildlife Service (USFWS https://www.fws.gov/) Cooperative Agreement #1448-40181-99-G, U.S. Navy (https://www.navy.mil/) Agreement #W9126G-16-2-0002. The funders had no role in study design, data collection and analysis, decision to publish, or preparation of the manuscript.

**Competing interests:** The authors have declared that no competing interests exist.

monitored [2] and should also be easily interpreted by restoration practitioners and managers [1]. The Florida Everglades is part of a larger regional watershed encompassing the Kissimmee-Okeechobee-Everglades-Florida Bay system at 1.5 million acres and is the largest subtropical wilderness in the United States and designated as a World Heritage Site and an International Biosphere Reserve [3].

Large-scale water control projects completed over the past 150 years have dramatically transformed the Florida Everglades ecosystem from a vast continuous subtropical wetland into a highly compartmentalized human-dominated system comprised of agricultural lands, urban landscape, and a large network of canals [3] constructed for flood risk reduction and water supply [4, 5]. Freshwater that once flowed south through the Everglades into southern estuaries and Florida and Biscayne bays have decreased and have been diverted resulting in altered spatial patterns of ecotones and salinity regimes throughout coastal wetlands [4, 6, 7]. There are more areas of Florida and Biscayne bays experiencing hypersalinity events ($\geq$ 40 psu) for longer periods of time [8, 9]. Changes in freshwater supply, including the length of hydroperiod, have reduced, degraded, or in some cases eliminated critical estuarine habitat necessary for aquatic communities [5, 6, 10, 11] and increased the probability of saltwater intrusion [9].

The Comprehensive Everglades Restoration Plan (CERP) tasked with restoring the hydrological function of the Everglades [12]; appropriately dubbed the "River of Grass" [13] is one of the world's largest and costliest ecosystem restoration projects [14], with an estimated $14.8 billion dollar budget [15, 16]. Restoration objectives established by CERP for the Everglades and associated estuaries are to increase the quantity and quality of freshwater supplied to Florida and Biscayne bays and to better coordinate the timing of freshwater delivery to estuaries to mitigate extreme conditions [17] (S1 Table).

The American crocodile (*Crocodylus acutus*) is the most widely distributed species of New World crocodiles [18] and occurs at its northernmost distribution in South Florida, then across coastal Mexico, down into South America and along the Caribbean [18]. The species has experienced severe declines due to overexploitation and loss of habitat for nesting throughout its historical range [19]; and is presently classified as Vulnerable on the International Union for Conservation of Nature (IUCN) Red List [18] and is on Appendix I of the Convention on International Trade in Endangered Species of Wild Fauna and Flora (CITES). In South Florida, nesting of American crocodiles was restricted to a small area of Northeastern Florida Bay (NEFB) in Everglades National Park (ENP) and Northern Key Largo by the early 1970s [20]. In 1975, the species was placed on the Federal Endangered Species List (Federal Register 40), but with critical monitoring and management efforts [21, 22], the Florida population of American crocodiles was reclassified from endangered to threatened in 2007 (Federal Register 72, [23]).

American crocodiles are considered an ecological indicator species in the Florida Everglades, because crocodile survival and population dynamics are directly connected with regional hydrological conditions [24, 25]. Crocodile responses are tightly linked to patterns of freshwater supply to southern estuaries that influence water depth, salinity regimes, and ultimately, resource availability [25–31].

Specific ecosystem restoration goals for crocodiles in Northeastern Florida Bay are to restore Taylor Slough as a main source of freshwater, to re-establish early dry season flow (October to January) from Taylor Slough to NE Florida Bay, and to re-establish a fluctuating mangrove backcountry salinity that rarely exceeds 20 psu [22] (Fig 1). American crocodiles do not have specialized physiological adaptations for a marine existence [32–34] and in laboratory studies, the species is able to grow in saline conditions when brackish water was available for drinking [34]. Hatchling crocodiles, however, were unable to maintain mass and further lost mass under hypersaline ($\geq$40 psu) conditions [20, 29, 34, 35]. In field studies, Kushlan and

Mazzotti [20] observed a distinct crocodile preference for habitats with fresh to brackish water averaging 14 ppt (parts per thousand sea water) with a seasonal shift to higher salinity habitats during nesting. Based on these studies, improved freshwater delivery in NE Florida Bay, including into mid-dry season, would decrease hatchling dispersal distance, stimulate food production, and improve crocodile relative density, body condition, growth, and probability of survival to maturity [22].

By explicitly linking population monitoring of indicator species to objectives of resource management, monitoring results can provide better understanding of ecosystem changes that can be used to evaluate management effectiveness and inform future policy development [30, 36]. Using long-term monitoring data, we evaluate short-term (body condition), intermediate (growth), and long-term (survival) responses of crocodiles to ecosystem restoration efforts in South Florida, particularly targeted at improved salinity conditions (see S1 Table for timeline of restoration efforts). We examine the hypothesis that in areas where hydrological conditions have been most disturbed, such as NE Florida Bay, where freshwater flow has been reduced and diverted and has resulted in hypersalinity conditions, negative effects will be most pronounced in American crocodiles.

## Methods

### Study site

Our study site is at the southern tip of Florida, USA, and is on the edge of the northernmost range of the American crocodile. Our site encompassed Northern and Southern Biscayne Bay, which is mainly within Biscayne National Park, south of Coral Gables Waterway, then south along Florida Bay and west to Cape Sable (Fig 1). The total area of Biscayne Bay is 1,110 km$^2$ with a depth < 1 m near shore. Florida Bay is 2,200 km$^2$ total area and is a shallow lagoonal estuary with an average depth < 1 m and receives freshwater from two major drainage basins, Shark River Slough and Taylor Slough [7]. Coastal Everglades National Park is connected to Florida Bay by way of many creeks, ponds, small bays, and a few engineered canals and ditches [26, 28].

There is less than half of the freshwater currently flowing into Florida Bay compared to pre-drainage conditions and salinity conditions in the bay reflect a long-term flow signal and a short-term rainfall signal [7, 14, 37]. More than 80% of Florida Bay is within the boundaries of Everglades National Park and serves as vital habitat for Florida wildlife [7]. National Park Service managers plugged canals in western Florida Bay, such as East Cape Canal, Buttonwood, Homestead, and Flamingo canals, with intent to reduce saltwater intrusion first in the late 1950s and early 1960s, then in 1986 and 1997 as plugs failed, and more recently in 2011 (S1 Table) [22, 38].

We grouped survey routes into hydrologically distinct areas based on proximity to each other and potential crocodile use: Crocodile Lake National Wildlife Refuge, North Key Largo, Barnes Sound, and Manatee Bay (CRL); NE Florida Bay from US1 to Alligator Bay (including Long, Little Blackwater and Blackwater Sounds, Joe Bay, Davis Cove, Deer Key, and Alligator Bay) (NEFB); Little Madeira Bay, Taylor River, and Madeira Bay (MADB); West, Cuthbert, Long, Seven Palm, Middle, and Monroe Lakes, and Terrapin Bay (WEST); North and South Biscayne Bay and Card Sound (BBC); Cape Sable beaches, East Cape Canal, Lake Ingraham and associated creeks (CAPE); Flamingo, Buttonwood and Homestead canals, Coot Bay, Mud and Bear Lakes (FLAM; Fig 1). Northeast Florida Bay was split into NEFB and MADB to detect any effects of freshwater flow from Taylor River (see Fig 1) that empties directly into Little Madeira Bay, however geographically Madeira Bay is contained within NEFB.

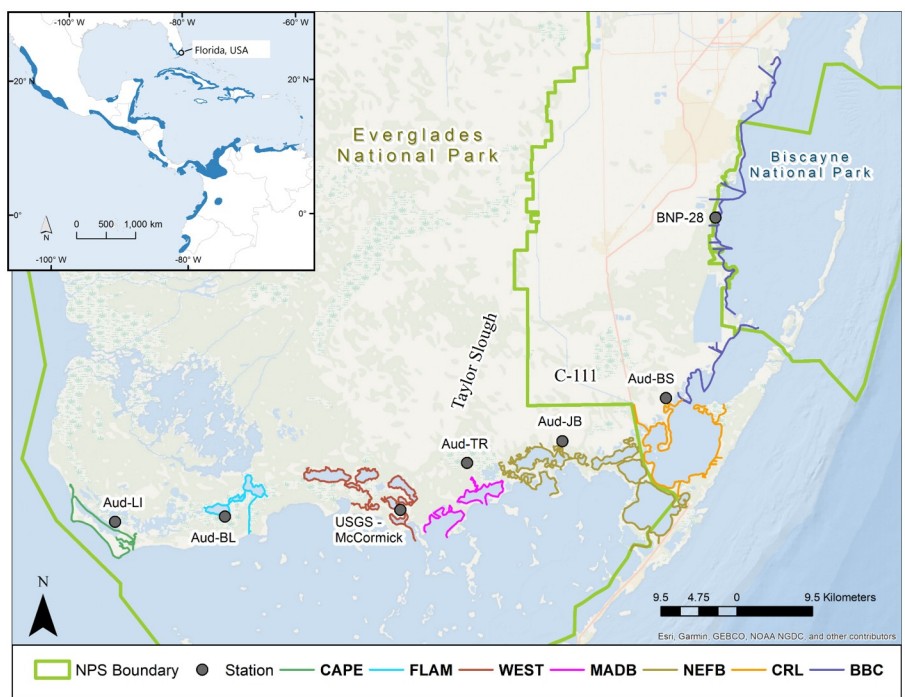

**Fig 1. Study area of American crocodile (*Crocodylus acutus*) survey routes from Biscayne Bay Complex west to Cape Sable in South Florida, USA.** Survey routes are grouped by areas: BBC = North and South Biscayne, and Card Sound; CRL = Crocodile Lake National Wildlife Refuge, North Key Largo, Barnes Sound, and Manatee Bay; NEFB = NE Florida Bay from US1 to Alligator Bay (including Long, Little Blackwater and Blackwater Sounds, Joe Bay, Davis Cove, Deer Key and Alligator Bay); MADB = Little Madeira Bay, Taylor River, and Madeira Bay; WEST = West, Cuthbert, Long, Seven Palm, Middle, and Monroe Lakes, and Terrapin Bay; FLAM = Flamingo, Buttonwood and Homestead canals, Coot Bay, Mud and Bear Lakes; CAPE = Cape Sable beaches, East Cape Canal, Lake Ingraham and associated creeks. Representative hydrological monitoring stations for each area are indicated with a black circle.

Shoreline vegetation along Florida Bay is primarily comprised of red mangrove (*Rhizophora mangle*), black mangrove (*Avicennia germinans*), white mangrove (*Laguncularia racemosa*), and buttonwoods (*Conocarpus erectus*). Australian pine (*Casuarina spp.*) and Brazilian pepper (*Schinus terebinthifolius*) dominate protected shorelines along canals and ponds in developed areas [39].

## Salinity data

We used mean daily salinity readings collected from stations managed by U.S. Geological Survey, ENP, Biscayne National Park, South Florida Water Management District, and Audubon Florida (Fig 1). Salinity data were consistently recorded at monitoring stations from 2000 onward. Representative stations were selected near crocodile survey locations as a proxy for environmental conditions in that immediate area. We used mean daily salinity from hourly readings to calculate annual minimum and maximum salinity, number of days above a high salinity threshold ($\geq 40$ psu = hypersalinity), and number of days below a low salinity threshold ($< 20$ psu) as parameters in regression analyses to investigate the relationship between salinity and crocodile indicators of body condition, growth, and survival.

## Population monitoring

We conducted nocturnal capture surveys by boat along accessible coastal and estuarine shorelines between East Cape at the western boundary of ENP, to south Biscayne Bay (Fig 1) from February 1978 to December 2015. From 1978–1996 surveys were conducted during the hatching (June -September) and post-hatching (September-November) periods [see 22]. Quarterly surveys were implemented between 1996 and 2009 (January-March, Quarter 1; April-June, Quarter 2; July-September, Quarter 3; October-December, Quarter 4). Budget cuts reduced surveys to Quarters 1, 2, and 4 in 2010 and 2011, and was further reduced to Quarters 1 and 4 in 2012 through 2015 [see 30]. Surveys were conducted only during appropriate environmental conditions (i.e., winds <15 knots, non-full moon nights and not at low tide [30, 40]; and consistent effort was made to capture crocodiles during surveys. Crocodiles were detected by the reflective layer in their eyes (*tapetum lucidum*), which produces a red, orange, or yellow "eyeshine" when illuminated by a spotlight.

Crocodiles were captured by hand, tongs, net, or noose and individually marked by notching caudal scutes according to a prescribed sequence [26]. We collected morphometric data, including total length (TL), snout-vent length (SVL), body mass, and determined sex when possible. We assigned crocodiles to size classes based on TL measurements, size classes are defined as follows: hatchling (TL < 65cm), juvenile (65 ≤ TL <150cm), subadult (150≤ TL < 225cm), and adult (TL ≥ 225cm) [26]. We categorized hatchling crocodiles based on time of year, differentiating between individuals observed within the hatching season (June–September) from those observed outside of the hatching season. We limited adult captures to between September 15th and March 15th, to minimize impacts on reproductive activities. We released crocodiles at the site of capture, and recorded date, time, location (measured by global positioning system, GPS), salinity (measured by an optical refractometer from 0–100 ppt) and habitat type (i.e., canal, cove, pond, creek, river, or exposed shoreline) for each crocodile capture event.

## Crocodilian biological parameters

**Body condition.** Body condition, a ratio of mass to length, is expected to be higher in large animals, but it is more an indication of relative fitness [41] and has been used to inform on how well a crocodilian is doing in its environment [31, 42–44]. Studies show body condition of crocodilians is affected by prey availability and diet [45, 46], habitat suitability and seasonal fluctuations of environmental conditions, including severe weather [31, 47–49]; growth [50, 51], and parasite loads [52].

To assess body condition, we calculated Fulton's condition factor (*K*) as follows:

$$K = 10^2 * \frac{W}{L^3} \qquad (1)$$

where *W* is body mass (g) and *L* is SVL (cm).

Data met assumptions of normality to calculate Fulton's K and we used multivariate linear regression analysis to investigate relationships of biotic and abiotic factors with body condition. Three models were compared: model parameters in the "basic" model included year and season (wet season: May to September, dry season: October to April), location of capture (i.e., NEFB, FLAM, CAPE, etc.), size class, and habitat type. The "salinity" model assessed the relationship between salinity and crocodile body condition, including minimum and maximum salinity, and number of days above and below salinity thresholds. The "location" model assessed the combined effects of capture location and salinity. We compared models based on

their Akaike Information Criteria (AIC), and selected the best model with lowest AIC as the most parsimonious.

**Growth rate modeling.** We first modeled the general shape of total length as a function of age using generalized linear models for growth curve analysis [see 53]. We calculated crocodile age as time between initial capture as a hatchling and the time of recapture (i.e., $t_1$-$t_0$). Using all crocodiles of known age captured from 1978 to 2015, we compared a constant model with three models including linear, quadratic, and cubic terms of age incrementally (i.e., first-, second-and third-order polynomials) on a (natural) log–log scale. We included longitude (easting) (to indicate physical location of capture) and its quadratic effect as additive effects on growth. After selecting the most parsimonious model for the general shape based on their lowest AIC, we investigated effects of salinity (minimum and maximum salinity, and average salinity during the wet and dry seasons), and location of capture on crocodile growth.

**Survival rate analyses.** We estimated age-specific survival rates based on capture-recapture data of crocodiles with known age (i.e. non estimated) as they were captured and marked as hatchlings. We performed capture-recapture analyses (CR; [54]). This analysis estimates two parameters in the survival rate model: estimated annual survival rate (proportion of crocodiles that survived between time $t$ and $t+1$, later referred to as Φ) and recapture rate at time $t$ (later referred to as $p$, [55, 56].

Most hatchlings (94%) were captured between June and September, peaking in July (S1 Fig); thus, we defined June 1st as the starting point (noted $t$) for each year in the CR analysis [54]. We first modeled constant, time- and age-dependent recapture rates. We then fit a full age-dependent model for survival because we had no *a priori* knowledge about the age-structure of crocodile survivorship. From the observed pattern of age-specific survival (survival between age $a$ and age $a+1$), we pooled age-classes with similar survival rates to reduce variation in survival rate estimates. This action resulted in four age-classes, defined as follows: 0–1 year, 1–2 years, 2–6 years, and $\geq$ 6 years. We then investigated the effects of average salinity, and area of capture by adding them as linear effects on estimated crocodile survival rates. We selected models with the smallest AIC as the best performing model.

All analyses were performed using R 3.6.0 [57], with the notable help of the packages RMark [58] for survival analysis (based on Program Mark software; [59], and ggplot [60], and cowplot [61] for graphs.

**Ethics statement.** Animal subjects were treated ethically, and research was conducted under U.S. Fish and Wildlife Service permit #TE077258-2 and adhered to welfare standards approved by the University of Florida IACUC #201509072 and University of Florida ARC #002-08FTL. The authors acknowledge that there are no conflicts of interest related to this article.

## Results

### Population monitoring

Across the study domain, a total of 10,040 crocodile capture events occurred between 1978 and 2015. Most captures (87%) occurred in three areas: FLAM (26%, N = 2785), CAPE (36%, N = 3900), and NEFB (25%, N = 2665). Captures were also made in WEST (5%, N = 542), BBC (3%, N = 354), and CRL areas (5%, N = 516). Size measurements were collected from a total 9,685 crocodiles, resulting in size data for 8,723 hatchlings, 465 juveniles, 271 subadults, and 226 adults. Number of crocodiles captured annually ranged from four individuals in 1981 to a maximum of 1,425 individuals in 2015 (Fig 2A). More than 90% of captures were of hatchlings during the summer hatching months from June to August (Fig 2A and 2B).

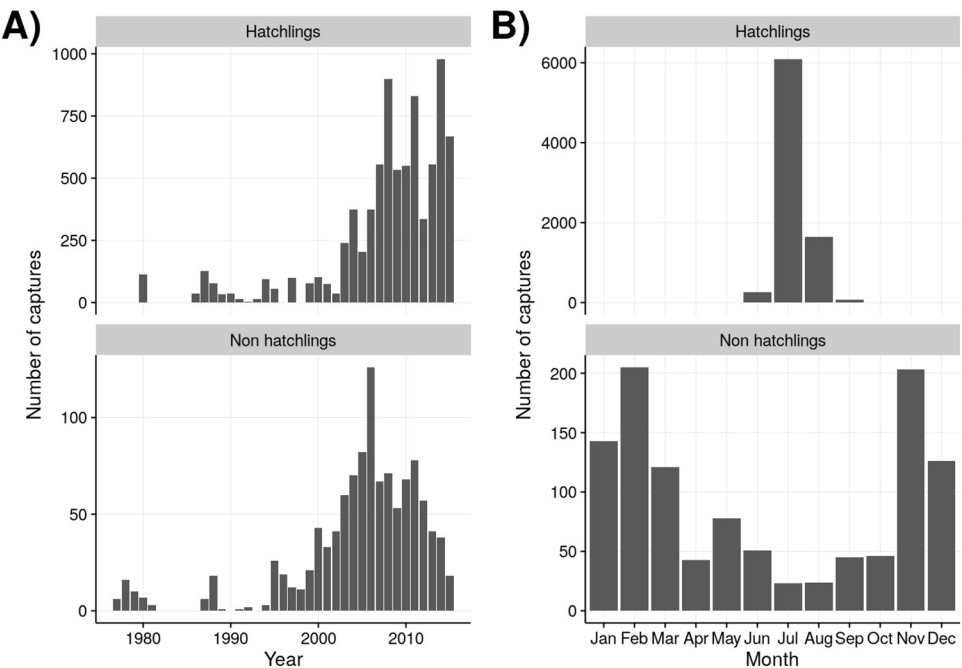

**Fig 2. A) Annual and B) monthly number of American crocodile (*Crocodylus acutus*) captures in South Florida between 1978 and 2015.** No surveys were conducted 1982–1985. Hatchling and non-hatchling crocodiles are separated, and scales differ based on the large number of hatchlings captured relative to non-hatchlings.

## Salinity

Salinity varied across years and among sites (Fig 3). The range of salinity in BBC and FLAM fluctuated between 15–30 psu and rarely exceeded 40 psu whereas, CAPE and CRL demonstrated more marine environments (marine = 35 psu) with much less fluctuation (CAPE mean 34.1 ± 9.0 SD psu, CRL mean 31.4 ± 8.9 SD psu). Salinity conditions for NEFB (mean 31.3 ± 10.7 SD psu) and MADB (mean 34.2 ± 11.9 SD psu) were more variable fluctuating from estuarine < 10 psu to hypersaline ≥40 psu (Fig 3A). BBC and WEST have intermediate salinity conditions less than 30 psu (Fig 3B). In 2011, managers initiated the C-111 Spreader Canal Restoration project intended to increase freshwater flow to the South Florida region, particularly eastern and central Florida Bay [12] (S1 Table). These preliminary efforts do not produce immediate changes in salinity but show a slight decrease in extreme hypersaline events three to four years after project initiation with less variability in salinity conditions (Fig 3A) across the region, including in NEFB, MADB and WEST, where restoration projects are targeted.

## Body condition

We calculated Fulton's *K* for 859 non-hatchling crocodiles, with 449 juveniles (52%), 261 subadults (31%), and 149 adults (17%). Mean crocodile body condition was 2.14 ± 0.35 SD for all size classes across the study period and though there was annual variation, condition scores were generally above 2.0. Low body conditions scores (below 2.0, based on reference quartiles developed for South Florida crocodiles [62] were estimated in 1994, 1997, and 1998 (Fig 4A).

The basic model for body condition including the effects of year, season, size class, area, and habitat type, explained 33% ($R^2$ = 0.33) of the variation in crocodile body condition.

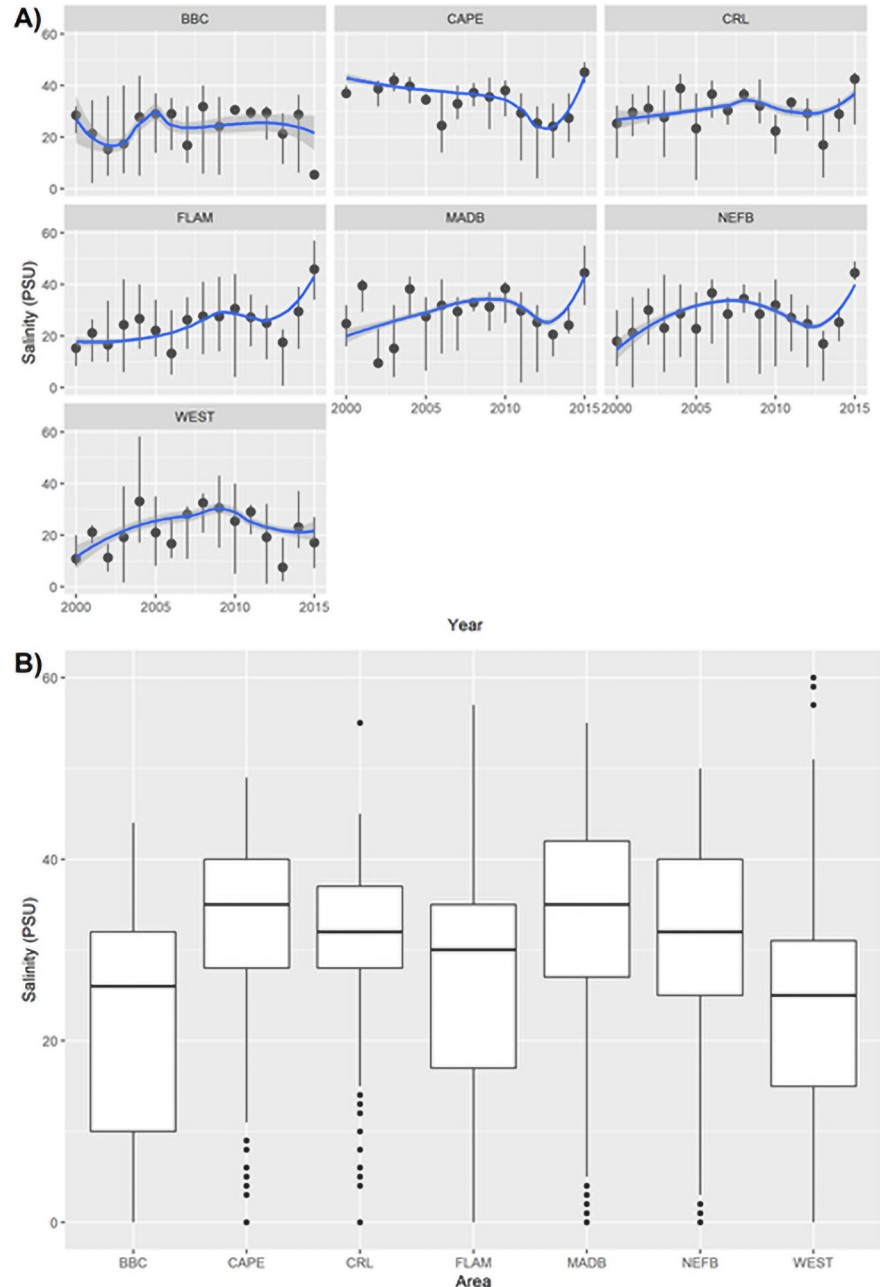

**Fig 3. A) Mean ±1 SD annual salinity measurements and B) Overall average salinity measurements for areas of South Florida taken from representative hydrological monitoring stations from 2000 to 2015.** Box plots represent median with 25% and 75% percentiles.

Season, size class, area, and habitat were significant predictors of body condition (all *P< 0.001*; Table 1), while year was found to be an uninformative predictor of body condition. Body condition was lowest in the juvenile size class and increased through ontogeny (Fig 4B, Table 1). We found crocodiles of all sizes classes captured in the wet season to be in poorer condition than those captured in the dry season (Fig 4B, Table 1).

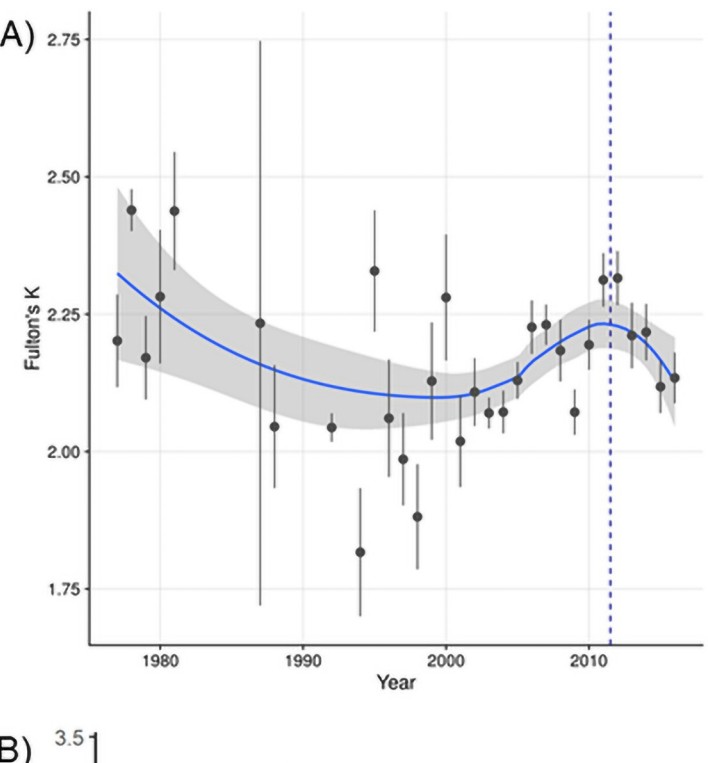

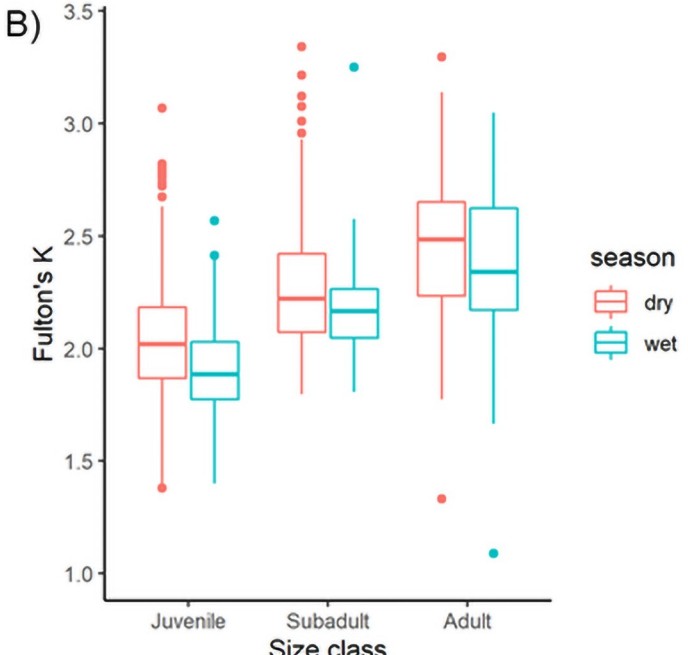

**Fig 4.** Fulton's K condition factor of American crocodiles (*Crocodylus acutus*) captured in South Florida A) between 1978 and 2015. Black circles are mean values and error bars represent ± 1SE, vertical dashed line marks before and after 2012 operation of the C-111 Spreader Canal Restoration Project; B). by size classes for each season. Box plots represent median with 25% and 75% percentiles.

**Table 1. Multivariate linear regression analysis of body condition of American crocodiles (*Crocodylus acutus*) captured in South Florida from 1978–2015.**

| Variable | β | SE | P |
|---|---|---|---|
| **Year** | -0.001 | 0.001 | 0.942 |
| **Season (wet/dry)** | -0.105 | 0.026 | <0.001*** |
| **Size class (ref Juvenile)** | | | |
| Size class Subadult | 0.304 | 0.024 | <0.001*** |
| Size class Adult | 0.453 | 0.029 | <0.001*** |
| **Area (ref NEFB)** | | | |
| Area CRL | 0.027 | 0.040 | 0.505 |
| Area WEST | 0.116 | 0.031 | <0.001*** |
| Area BBC | 0.102 | 0.041 | 0.014** |
| Area CAPE | 0.098 | 0.040 | 0.015** |
| Area FLAM | 0.246 | 0.034 | <0.001*** |
| **Habitat (ref Canal)** | | | |
| Cove | -0.001 | 0.030 | 0.963 |
| Pond | 0.133 | 0.052 | 0.011** |
| Creek/River | 0.076 | 0.043 | 0.076 |
| Exposed Shoreline | 0.090 | 0.059 | 0.131 |
| ***Salinity Model (N = 707)*** | | | |
| **Size class (ref Juvenile)** | | | |
| Size Class Subadult | 0.294 | 0.027 | <0.001*** |
| Size Class Adult | 0.452 | 0.032 | <0.001*** |
| **Salinity Days Low/year** | -0.001 | 0.001 | 0.056 |
| **Salinity Days High/year** | -0.002 | 0.001 | 0.009** |
| **Salinity Max/year** | -0.002 | 0.002 | 0.402 |
| **Salinity Min/year** | 0.010 | 0.003 | 0.002** |
| ***Salinity and Area Model (N = 707)*** | | | |
| **Size class (ref Juvenile)** | | | |
| Size class Subadult | 0.311 | 0.028 | <0.001*** |
| Size class Adult | 0.467 | 0.034 | <0.001*** |
| **Area (ref NEFB)** | | | |
| Area CRL | 0.101 | 0.049 | 0.833 |
| Area WEST | 0.103 | 0.043 | 0.018** |
| Area BBC | 0.079 | 0.059 | 0.179 |
| Area CAPE | 0.172 | 0.080 | 0.031* |
| Area FLAM | 0.229 | 0.065 | <0.001*** |
| **Salinity Days Low/year** | -0.001 | 0.001 | 0.128 |
| **Salinity Days High/year** | -0.001 | 0.001 | 0.723 |
| **Salinity Max/year** | -0.003 | 0.002 | 0.248 |
| **Salinity Min/year** | -0.002 | 0.005 | 0.677 |

Salinity is measured in practical salinity units, β is an unstandardized coefficient of regression, Area/Location (NEFL = NE Florida Bay area, CRL = Crocodile Lake National Wildlife Refuge area, West = West Lake/Seven Palm area, BBC = Biscayne Bay Complex, Cape = Cape Sable area, Flamingo = Flamingo area,

*$p < 0.05$,

**$p < 0.01$,

***$p < 0.001$.

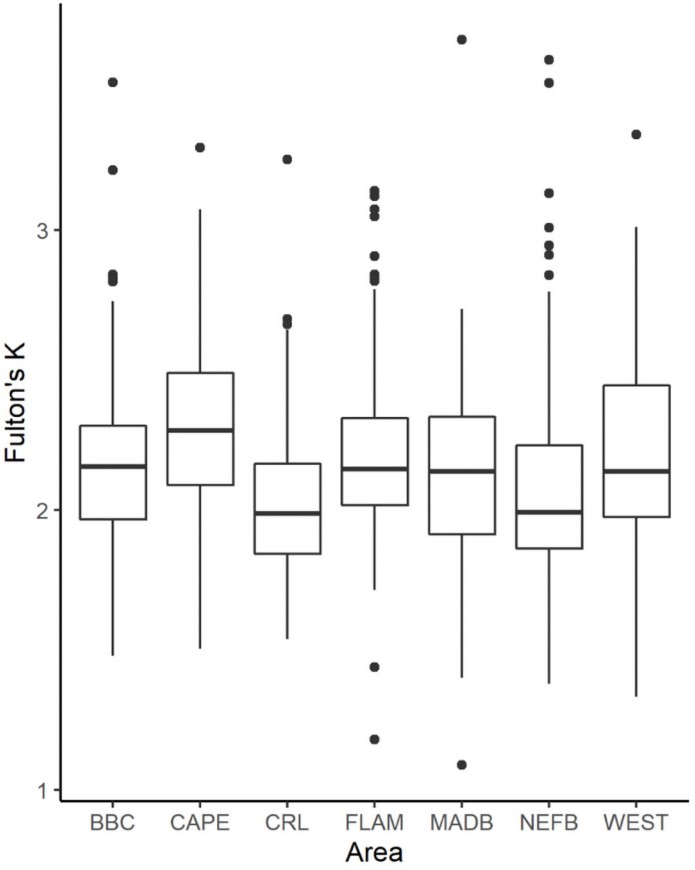

**Fig 5. Fulton's K condition factor of American crocodiles (*Crocodylus acutus*) captured in South Florida between 1978 and 2015 by capture area.** Black circles are mean values and error bars represent ± 1SE. Box plots represent median with 25% and 75% percentiles.

Among areas, body condition was lower in NEFB (mean = 2.07 ± 0.35 SD) and CRL (mean = 2.03 ± 0.29 SD) relative to elsewhere in South Florida and crocodiles captured in CAPE maintained the highest body condition (mean = 2.30 ± 0.33 SD, Fig 5, Table 1). Individuals captured in ponds were in better condition than those in canals (Table 1). Crocodile body condition has been variable over time within ENP (Fig 6), and crocodiles from NEFB exhibited the most variability in body condition with scores lower than 2.0 for several years, but on average having a body condition slightly above 2.0. Crocodiles from FLAM and CAPE had an average body condition index above 2.25 but has decreased since 2010 and is approaching body condition scores similar to NEFB (Fig 6). When sex was included, sample size decreased and model fit was greatly reduced ($R^2$ = 0.19); however, sex was found to be a significant predictor of body condition with males exhibiting significantly poorer body condition than females ($P$ = 0.04).

The salinity model included size class, number of days ≥40 psu, number of days below 20 psu, annual minimum and maximum salinity; and explained 29% of variability in crocodile body condition. Size class and low annual salinity had positive effects on body condition (Table 1). Crocodiles exhibited poorer body condition when exposed to a dry season that had an average salinity of at least 37.5 psu and body condition decreased with more days spent in

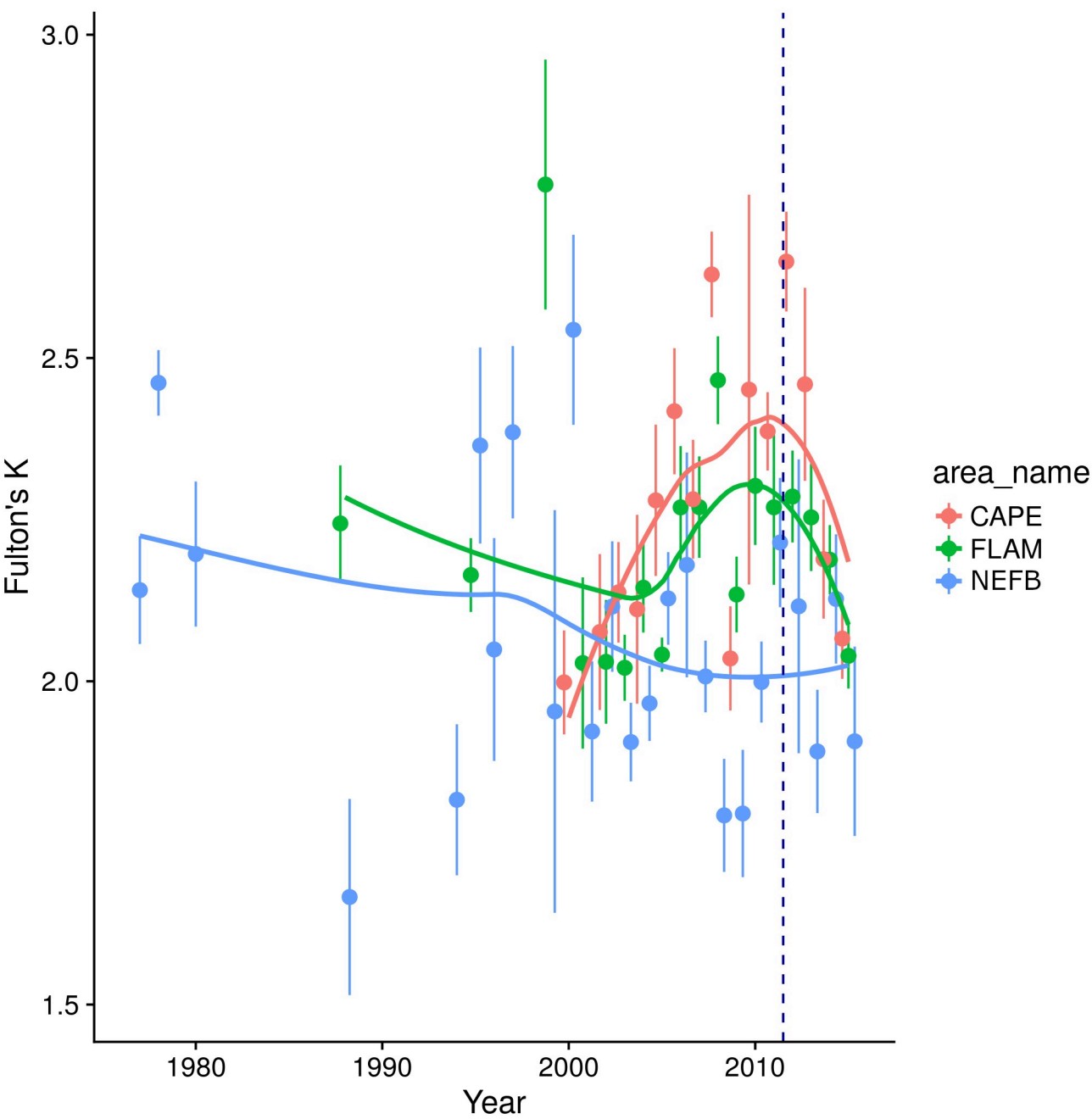

**Fig 6. Fulton's K body condition factor of American crocodiles (*Crocodylus acutus*) captured in South Florida between 1978 and 2015.** Areas included within Everglades National Park (CAPE = Cape Sable, FLAM = Flamingo area, NEFB = NE Florida Bay). Filled circles represent mean values and error bars extend to ± 1SE, dashed line at 2012 indicates start of C-111 Spreader Canal Restoration Project.

hypersaline conditions (≥40 psu, Table 1). When area was included as an additive effect to the salinity model, the model improved and explained 32% of the variability in body condition. Area and size class were the only significant predictors of body condition and accounted for the variation in salinity effects (Table 1).

**Table 2. Model selection table for growth analysis of American crocodiles (*Crocodylus acutus*) in South Florida.**

| Model | K | AIC | ΔAIC$_c$ | ω$i$ |
|---|---|---|---|---|
| *Global* | | | | |
| **Constant** | 2 | 930.53 | 1278.98 | 1.86$^{e-278}$ |
| **First order polynomial** | 3 | 174.28 | 522.73 | 3.07$^{e-114}$ |
| **Second order polynomial** | 4 | -338.89 | 9.57 | 8.30$^{e-03}$ |
| **Third order polynomial** | 5 | -348.46 | 0.00 | 9.92$^{e-01*}$ |
| **Sex** | | | | |
| **Constant** | 5 | -15.26 | 4.14 | 0.11 |
| **First order polynomial** | 7 | -13.02 | 6.38 | 0.04 |
| **Second order polynomial** | 9 | -19.40 | 0.00 | 0.86* |
| **Salinity** | | | | |
| **Constant** | 5 | -330.67 | 76.86 | 2.04 |
| **First order polynomial** | 9 | -386.29 | 21.24 | 2.45 |
| **Second order polynomial** | 21 | -407.53 | 0.00 | 9.99* |
| **Longitude** | | | | |
| **Constant** | 5 | -320.84 | 74.85 | 5.57 |
| **First order polynomial** | 7 | -348.18 | 47.52 | 4.81 |
| **Second order polynomial** | 13 | -395.70 | 0.00 | 1.00* |

Included models: global model, additive terms to the global model are sex, salinity, and longitude (easting). *K* is the number of parameters in the model; AIC is the Akaike information criterion; ΔAIC$_c$ is the difference of each model relative to the best model; ω$_i$ is the weight of evidence that each model is the best;

* model selected.

## Growth rate

We estimated crocodile growth rates using data from 573 captures of 376 individual crocodiles captured between 1978 and 2015. The best performing growth model included a cubic term of age (ΔAIC$_c$ = 0.00, Table 2), this model explained 89% (pseudo-R$^2$ = 0.89) of the variability in crocodile growth and was followed by the second order polynomial (ΔAIC$_c$ = 9.57) and out-performed all other models (Table 2). Growth was fastest for younger crocodiles within the first two years of life (Fig 7). Growth rate slowed with age; however, older crocodiles continued to grow and did not reach a plateau (Fig 7). Using a subset of data from 39 individuals of known sex (22 females, 17 males), we included sex as an additive term, and in interaction with all three polynomials (Table 2). Female and male crocodiles grew at different rates (S1 Fig). Females were 6.6% larger than males at 1 year old, and 13.3% larger at 5 years old, thus females grew faster than males in the earlier years of life. By 10 years of age, males were 18.3% larger than females (S1 Fig).

Salinity data were extracted from 2000 to 2015 for 505 crocodile captures representing 329 individuals. The best performing growth rate model included average salinity conditions during the dry season and explained 91% (pseudo-R$^2$ = 0.91) of variation in crocodile growth rates (ΔAIC$_c$ = 0.00, Table 2, Fig 3) and was better than first order polynomial (ΔAIC$_c$ = 21.24) and the constant models (ΔAIC$_c$ = 76.86). Variation in growth rates by capture location was accounted for in the salinity model (Table 2). Crocodiles exhibited different growth rates based on where they were captured (i.e., longitude). Crocodiles in FLAM, CAPE, and WEST areas grew faster than those in NEFB (S2 Fig). The growth model predicted a 13% decrease in growth after hatchling crocodiles were exposed to average dry season salinity conditions of at

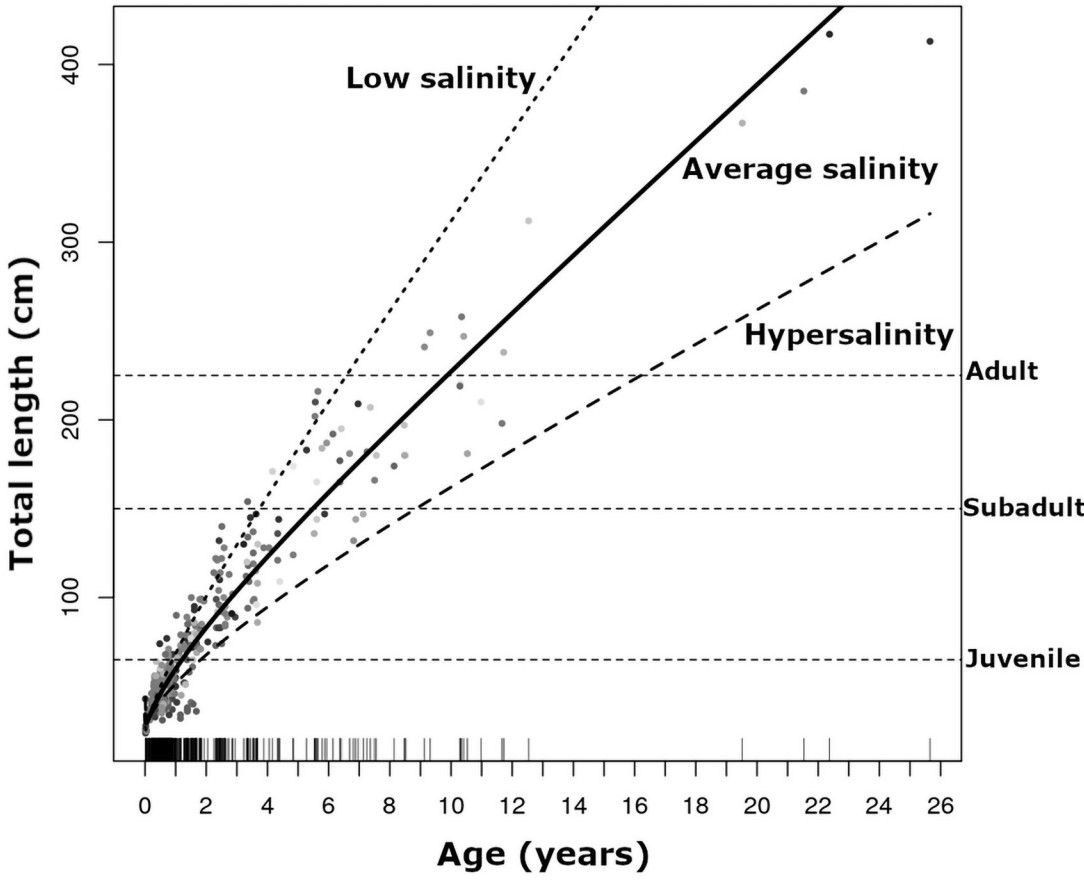

**Fig 7. Growth curves developed for American crocodiles (*Crocodylus acutus*) recaptured between 1978–2015 in response to salinity conditions in South Florida.** Horizontal dashed lines represent size classes.

least 37.5 psu in the first year. Under these dry season conditions, growth decreased by 24% after 5 years of exposure and by 29% after 10 years (Fig 3).

## Survival

A total of 9,040 crocodiles were initially captured as hatchlings; 1.5% (n = 132) of these survived at least one year. Mean time between recaptures was 474.4 days (~1.3 years) and the oldest recaptured crocodile was 22 years old. We used CR models to assess effects of age and time on apparent survival ($\Phi$) and recapture rate ($p$), as well as to investigate the age-structure of survival. We estimated mean hatchling survival to be 25% in South Florida. After one year of age survival increases to 41% and nearly doubles in the next three years. By age six, at the subadult stage, survival is close to 90% and remains consistently high into adulthood (Fig 8). Recapture rates were near zero prior to 1995 but increased with implementation of systematic monitoring surveys beginning in 2004 (S3 Fig). Hatchling survival rates were also highly variable across years and fluctuated between 15–70% (S4 Fig). The best performing model for crocodile survival ($\Phi$) included the fixed effects of area and age-class ($\Delta AIC_c = 0.00$, Table 3). The salinity model alone did not produce significant effects on survival ($\Delta AIC_c = 23.25$) but survival rates differed between areas (Table 3). Crocodiles from CRL and FLAM areas had the

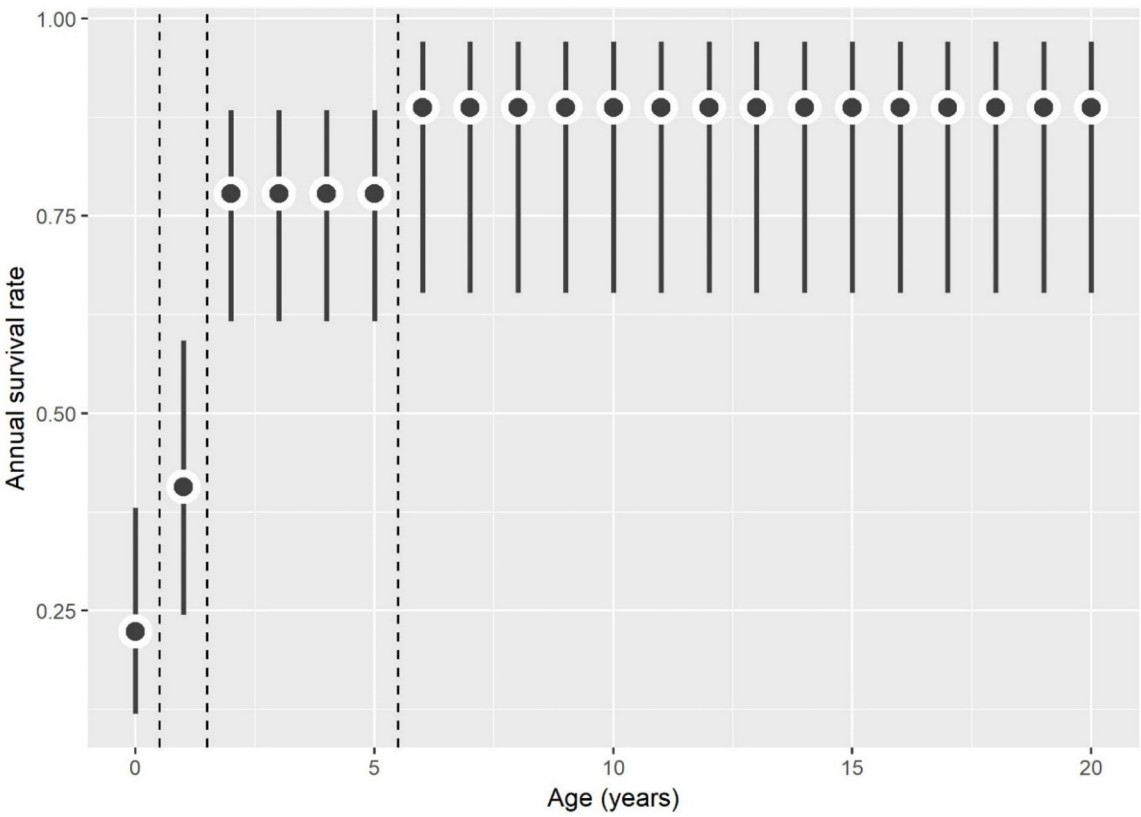

**Fig 8. Annual survival rates of American crocodiles (*Crocodylus acutus*) captured in South Florida as a function of age.** Dots represent mean values and error bars indicate 95% confidence intervals, dashed lines are age groups demonstrating similar survival rates (0–1 yr old, 1–2 yr old, 3–6 yr old, 7–22 yr old).

highest hatchling survival estimates of 69% and 53%. BBC had a 48% survival rate, and NEFB and MADB had lower hatchling survival rates of 34% and 31%. CAPE is a more recent nesting location and did not have the same 40-year period of nesting as other areas in South Florida and had a 28% hatchling survival rate.

**Table 3. Model selection for survival probabilities (Φ) of American crocodiles (*Crocodylus acutus*) in South Florida.**

| Model | K | Deviance | ΔAIC$_c$ | ω$_i$ |
|---|---|---|---|---|
| *Phi*(a_cl+area), p(cap a_cl) | 12 | 424.48 | 0.000 | 8.03$^{e-01}$* |
| *Phi*(a_cl+sal:area), p(cap a_cl) | 14 | 1446.67 | 1474.73 | 1.97$^{e-01}$ |
| *Phi*(a_cl), p(cap a_cl) | 6 | 455.95 | 19.44 | 4.83$^{e-05}$ |
| *Phi*(a_cl+sal), p(cap a_cl) | 8 | 1479.15 | 23.25 | 7.16$^{e-06}$ |
| *Phi*(a_cl+sal+I(sal$^{\wedge 2}$)), p(cap a_cl) | 8 | 1479.15 | 23.25 | 7.16$^{e-06}$ |

K is the number of parameters in the model, deviance relative to the best model, ΔAIC$_c$ is the difference of each model relative to the best model according to their Akaike Information Criterion corrected for a small sample size, ω$_i$ is the weight of evidence that each model is the best; a_cl = age classes; cap = crocodile captures, area = area crocodile was captured, sal = salinity;

*model selected.

## Discussion

The American crocodile is a flagship species of the Florida Everglades and this long-term monitoring project was initiated to track the once endangered, now listed as threatened South Florida population where there have been significant human-caused changes to the natural ecosystem [22, 25, 63]. From nearly 40 years of capture data, we assess in this study the body condition and additional population dynamics of growth and survival of American crocodiles, while Mazzotti et al. [30] assessed relative density. Our results show that hypersalinity conditions negatively affect these parameters and that where a crocodile is captured matters to its overall body condition, how fast it can grow, and ultimately to its survival.

In this study, mean crocodile body condition score was 2.14 and is acceptable crocodile body condition based on quartiles developed for South Florida crocodiles from 1978–2018 (ideal at K > 2.4, acceptable for K ≥ 2.0, and poor for K< 2.0, [62], similar to condition scores of *C. acutus* in Mexico of 1.75 to 2.642 [31]. Condition scores also fell within the ranges of other crocodilians i.e., *Crocodylus moreletii* in Belize of 2.03–2.70 [49], 1.88–2.45 for American alligators in the Florida Everglades (*Alligator mississipiensis*, [44], 2.16–2.39 for *Caiman crocodilus* and 2.30 for *Melanosuchus niger* in Brazil [64]; however we captured crocodiles lower than the regional average and less than 2.0 in NEFB, while CAPE crocodiles were in the best body condition. Directly influenced by resource availability, metabolic demands [65], and environmental conditions [46–48], body condition can have long-term impacts on reproductive readiness, clutch size, and number of nests [44, 66, 67].

As a large, long-lived reptile that demonstrates relatively slow body growth the American crocodile may express differential responses to ecological and environmental stressors at various life stages due to the inherent biological and physiological differences characteristic of each life-stage [53]. Growth rate, an intermediate-term response, integrates not only the conditions experienced at the time of hatching, but also throughout ontogeny. Here, growth rate was fastest in younger crocodiles (< 2 years old). The cumulative effects of environmental conditions and resource availability during the first dry season following hatching is highly critical to hatchling growth and survival. In lab studies, hatchling crocodiles exposed to hypersalinity (≥40 psu) conditions prior to reaching 200g body mass (40–45 cm TL), typically 3–4 months post-hatching experienced significantly lower growth rates and suffered a growth disadvantage that continued into adulthood [20, 29, 34, 35]. With increased hypersalinity exposure, crocodiles were likely to be more osmotically-stressed than those at low salinity conditions which translates into reduced growth and reductions in mass [54, 55]. Unlike many large-bodied animals that stop growing at adulthood and only increase in mass [68, 69], American crocodiles in South Florida exhibited continual (albeit slow) growth in length, and crocodiles continued to increase in body mass throughout life including for the oldest crocodile recaptured at 22 years of age.

In South Florida, survival of the American crocodile rests on juvenile survival in years leading up to adulthood [53]. In more recent years, there has been an increased threat of nest and hatchling predation by both Burmese pythons (*Python bivittatus*) and Argentine black and white tegus (*Tupinambis merianae*) as they have encroached on crocodile nesting sites [70]. Initially hatchling crocodiles, like young of other reptiles, exhibit high mortality rates and low survival [71]. Our estimated 25% annual survival rate for South Florida was within previous estimates from Key Largo, FL (7–43% [29]), and used in simulations [72], but is much higher than previously reported in Florida Bay (10% [73]), in ENP (estimated 10–25% [74]), at Turkey Point (9% [21], and 16% [53]), and in Panama (5% [75]). This survival rate also exceeds rates typical for other crocodilian species (6% in Indian gharials (*Gavialis gangeticus*) [76]; 5% in

Nile crocodiles (*Crocodylus niloticus*) [77]; 8% in Australian freshwater crocodiles (*Crocodylus johnstoni)* [78].

Newly recruited juveniles (>200g, > 65 cm) continue to grow most rapidly in lower salinity while becoming more tolerant of high salinities and less susceptible to predation [29] until they reach another critical milestone surpassing 75 cm TL, typically between 15–20 months of age in ENP [26, 28]. Once juveniles reach this size, predation risk is greatly reduced, and survival rates increase to 80% [this study; 53]. Moler [29] reported a similar second year survival of 64.9% in South Florida and is likely driven by increased dispersal ability and decreased predation risk [20]. By age six, subadult crocodiles have a 90% survival rate that continues into adulthood. The first few years of a crocodile's life are critical and are directly related to experiences during the first six months of life [26] and may be an indicator of environmental conditions [79]. Survival rates were highest in CRL (69%) and FLAM (53%) and much lower in NEFB (34%) and MADB (31%) areas. CAPE area is a more recent nesting colony and has a shorter period of crocodile presence (from 2000 onward) relative to historic NEFB (1978 onward), therefore there are fewer historical capture/recapture records to inform survival estimates.

Perhaps of equal importance in this study is that where a crocodile was captured had far-reaching effects on a crocodile's life. Crocodiles in NEFB exhibited the lowest body condition, slowest growth rate and lowest survival rates of anywhere in South Florida. In several species, there may be a trade-off between growth and body condition that enables individuals to reach adulthood faster at the cost of lower body condition [51, 80, 81], however crocodiles in NEFB were at disadvantages in each parameter measured in this study. When resources are limited, and prey are dispersed during the wet season, energy is typically allocated to one function (i.e. survival, reproduction, or growth) and reduces the amount of surplus energy available for storage [82]. Reduced prey availability in NEFB [83, 84] coupled with salinity conditions that fluctuate greatly and where hypersaline events last for the longest period of time contribute to suboptimal crocodilian habitat [22]. For example, at Joe Bay within NEFB for the 1989–1990 dry season there were 171 consecutive days above 35 psu and 141 of those days at ≥ 40 psu. Concurrently, at Taylor River (MADB), there were 125 days above 35 psu, with 117 days above 40 psu (these data). In several years following, salinity conditions above 35 psu persisted on an average of 42 consecutive days at Joe Bay and 47 days at Taylor River during the dry season. NEFB was historically under estuarine conditions (< 20psu) [9] but present-day conditions include increased salinization zones [37], altered hydroperiods [85], as well as saltwater intrusion further into interior estuaries that provide critical habitat for hatchling survival and vital prey source [22, 26].

In a seasonally pulsed ecosystem such as the Everglades, dry down periods are crucial for crocodilians [34, 35, 44] and nesting wading birds for reliable food sources, and during lengthy hydroperiods there is reduced prey availability [86–89]. Conversely, extreme dry downs also result in negative effects in crocodilians including lower body condition, reduced growth and decreased survival under hypersaline conditions. Nesting birds, as well as other vertebrates, also show negative responses to high salinity conditions in Florida Bay, so much so that entire wading bird colonies are greatly reduced and abandon historical nesting sites during extreme dry down events when water is both very low and salinity is high [83, 85, 89, 90]. Furthermore, high salt diets as a result of foraging in a marine environment, has been shown to reduce growth in nestling laughing gulls (*Larus atricilla*) [91] and nestling white ibis (*Eudocimus albus)* [92]; illustrating the dependence on inland foraging opportunities by wading birds to maintain health and survival.

Another short term-response and a performance measure for goals of CERP is crocodile relative density. Mazzotti et al. [30] used 12 years of crocodile monitoring data to investigate

the influence of salinity on relative density of crocodiles in ENP and found a relative density of 2.9 individuals/km and decreasing with increasing salinity. More specifically, crocodile relative density was greater in the FLAM and CAPE areas when compared to lower densities recorded in WEST and NEFB areas [30]. Salinity effects on body condition were diluted by area effects, suggesting that factors other than salinity affect body conditions (this study) and relative density of crocodiles [30]. For example, Mazzotti [26] found in NEFB that most sightings of crocodiles in higher salinities were females at nest sites, Rosenblatt and Heithaus [93] found that alligators moved to access higher prey abundance in full-strength seawater at the expense of exposure, Evert [94] found that relative density of alligators in Florida lakes was related to nutrient levels and Brandt et al. [44] found body condition of alligators in Florida lakes was related to nutrient status. We hypothesize that differences in nutrient levels among areas leads to differences in prey abundance which would affect both relative density and body condition. Relating occurrence of crocodiles to nutrient levels, and to distribution and relative density of prey items should improve our understanding of how crocodiles will respond to ecosystem changes.

Crocodiles in CAPE and FLAM areas maintained the highest body condition and fastest growth rates, while survival rates were also much higher in FLAM and CRL areas. These results are likely because of the plugging of canals at Cape Sable to reduce saltwater intrusion and of the ability for crocodiles to travel up into protected estuarine habitat in the nearby Fox Lakes for foraging [79]. Low crocodile hatchling survival has been correlated with greater travel distances to freshwater sources, low prey availability, and fewer refuge habitats [22, 28, 29]. This supports the hypothesis that freshwater diversion and the associated deterioration of habitat conditions in NEFB, an area that has historically supported high crocodile numbers, has negatively impacted the health and abundance of crocodiles there [22, 28, 30]. Crocodile numbers have increased in FLAM and CAPE areas [22, 79], which is likely due to the cumulative effects of ongoing restoration efforts near these areas (Table 1). The plug at Buttonwood Canal may have reduced saltwater intrusion northward and improved nursery habitat in this area (Table 1). In NEFB, poor nursery habitat was due to diversion of freshwater from Taylor Slough to the C-111 Canal. The Completed C-111 Spreader Canal restoration project is expected to restore a more natural water flow to Taylor Slough and to estuaries in NEFB [12] (Table 1). CERP objectives for Florida Bay are to reduce the number of hypersaline events that occur annually, as well as increase both the frequency and spatial distribution of low salinity conditions throughout the bay and minimize extreme events to establish more stable salinity conditions [95].

At each response interval (short-body condition, intermediate-growth, and long-term-survival), salinity had negative effects on crocodile indicator parameters. An adequate supply of freshwater is necessary to maintain good health and increase survival of crocodiles in South Florida, and where freshwater is limited or where nursery habitat is further from nesting habitat, there is additional stress imposed on individuals [22, 63, 96–98]. Our results show that crocodiles are effective ecological indicators of ecosystem responses to restoration of Everglades estuaries and underscores the need for continued long-term research to evaluate restoration progress. These findings emphasize the need to continue monitoring of NE Florida Bay where freshwater flow and salinity patterns are currently the target of restoration efforts [15, 16] and to evaluate outcomes of Cape Sable and Flamingo areas to inform resource managers.

## Supporting information

**S1 Fig. Total body length (TL) as a function of age of male and female American crocodiles (*Crocodylus acutus*) recaptured between 1978–2015 in South Florida.** Males are represented

as red circles with a Loess best-fit line in red and females are represented as black circles with a Loess best-fit line in black. Horizontal dashed lines represent size classes.
(TIF)

**S2 Fig. Average crocodile growth of *Crocodylus acutus* in South Florida in the first five years by area of capture.** Solid line represents average growth in West Lake and Seven Palm (7Palm) areas, dotted line is average growth rate at Flamingo and Cape Sable areas, and dashed line is average growth rates in NE Florida Bay. Horizontal dashed lines represent size classes within first five years: Juvenile and subadult.
(TIF)

**S3 Fig. Recapture rates of American crocodiles (*Crocodylus acutus*) in South Florida from 1978–2015.** Dots represent mean values and lines indicate 95% confidence intervals, dashed lines represent different phases of similar recapture rates (before 1995, 1995–2006, 2007–2015).
(TIF)

**S4 Fig. Annual hatchling survival rate of American crocodiles (*Crocodylus acutus*) in South Florida from 1978–2015.** Dots represent mean values and error bars are 95% confidence intervals, dashed lines reflect phases of recapture rate.
(TIF)

**S1 Table. Timeline of restoration events in the Florida Everglades.**
(DOCX)

## Acknowledgments

We thank present and previous members of The Croc Docs (http://crocdoc.ifas.ufl.edu/) for fieldwork, data collection and dedication to Everglades conservation, S. Picardi for producing the survival figure, S. Farris and D. Bucklin for producing Fig 1, C. Bonenfant for analytical assistance, S. Acacia Gonzalez for the opportunity to work on this manuscript, and M. Parry, S. C. Gonzalez, F. Briggs, and members of the CSG Argentina meeting for valuable comments. Any use of trade, firm, or product names is for descriptive purposes only and does not imply endorsement by the U.S. Government.

## Author Contributions

**Conceptualization:** Venetia S. Briggs-Gonzalez, Frank J. Mazzotti.

**Data curation:** Venetia S. Briggs-Gonzalez, Michael S. Cherkiss.

**Formal analysis:** Venetia S. Briggs-Gonzalez, Mathieu Basille.

**Funding acquisition:** Frank J. Mazzotti.

**Investigation:** Michael S. Cherkiss, Frank J. Mazzotti.

**Methodology:** Venetia S. Briggs-Gonzalez, Mathieu Basille, Frank J. Mazzotti.

**Project administration:** Michael S. Cherkiss, Frank J. Mazzotti.

**Resources:** Frank J. Mazzotti.

**Software:** Mathieu Basille.

**Supervision:** Frank J. Mazzotti.

**Visualization:** Venetia S. Briggs-Gonzalez, Mathieu Basille, Michael S. Cherkiss.

**Writing – original draft:** Venetia S. Briggs-Gonzalez, Mathieu Basille.

**Writing – review & editing:** Venetia S. Briggs-Gonzalez, Mathieu Basille, Michael S. Cherkiss, Frank J. Mazzotti.

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
