## [Decision Letter · Decision Letter 0]

18 Dec 2020

PONE-D-20-34204

American crocodiles (Crocodylus acutus) as restoration bioindicators in the Florida Everglades

PLOS ONE

Dear Dr. Briggs-Gonzalez,

Thank you for submitting your manuscript to PLOS ONE. Your manuscript was assessed by 3 subject experts and myself. All three of us agree that the manuscript is interesting and generally well prepared and presented. However, after careful consideration, we feel that the work as presented does not fully meet PLOS ONE’s publication criteria. Therefore, we invite you to submit a revised version of the manuscript that addresses the points raised during the review process.

Required changes:

1. Please address the comments of Reviewer 1 and the AE, who both find that the manuscript is written under the assumption that readers have some familiarity with south Florida and the Everglades. PlosOne is an international journal with a diverse readership, so some more general information is required. For example, Figure 1 showing the study site should provide a larger geographic context, and the text-based description should include some more general location and site information. This need not be extensive, a few sentence will do.

2. Please address the comments and concerns raised by Reviewer 1 regarding data management and statistical analyses. Most of these appear to stem from a lack of detail provided in the manuscript, and may only require changes to the text.

3. Please balance the presentation of data vs. the output from statistical modeling. As presented the Results in main manuscript deal almost exclusively with tables of model outputs, and the actual data displays are relegated to the Supplemental Material. Figures 1-4 in the Supplemental Material should likely be part of the main document. In contrast, Table 1 is less useful as part of the main document and may be better moved to the SI section.

Recommended Changes:

1. Please carefully read and consider the comments provided by the 3 subject experts and the AE. All of these are meant to improve the presentation of the manuscript. Make changes or respond as appropriate,

We look forward to receiving your revised manuscript.

Kind regards,

Christopher M. Somers

Academic Editor

PLOS ONE

Journal Requirements:

2.) We note that you have stated that you will provide repository information for your data at acceptance. Should your manuscript be accepted for publication, we will hold it until you provide the relevant accession numbers or DOIs necessary to access your data. If you wish to make changes to your Data Availability statement, please describe these changes in your cover letter and we will update your Data Availability statement to reflect the information you provide.

3.) Please include a caption for figure 1, 2, 3 and 4.

4.) We note that Figure 1 in your submission contain map images which may be copyrighted. All PLOS content is published under the Creative Commons Attribution License (CC BY 4.0), which means that the manuscript, images, and Supporting Information files will be freely available online, and any third party is permitted to access, download, copy, distribute, and use these materials in any way, even commercially, with proper attribution. For these reasons, we cannot publish previously copyrighted maps or satellite images created using proprietary data, such as Google software (Google Maps, Street View, and Earth). For more information, see our copyright guidelines: http://journals.plos.org/plosone/s/licenses-and-copyright.

5.) Please include captions for your Supporting Information files at the end of your manuscript, and update any in-text citations to match accordingly. Please see our Supporting Information guidelines for more information: http://journals.plos.org/plosone/s/supporting-information.

See "Required Changes" above.

Additional Editor Comments (if provided):

Table – contains a lot of information but is only referred to in passing in the text. It deserves more coverage, or should be removed / placed in SOM.

Figure 1 – needs to show larger geographic context for general and international readership.

No mention of invasive species and changes in competition levels? E.g., Burmese pythons? This kight be relevant in the Discussion

Consider adding SI figs 1-4 to the main manuscript to balance presentation of data with model tables.

Reviewers' comments:

Reviewer's Responses to Questions

**Comments to the Author**

1. Is the manuscript technically sound, and do the data support the conclusions?

Reviewer #1: Yes

Reviewer #2: Yes

Reviewer #3: Yes

2. Has the statistical analysis been performed appropriately and rigorously? 

Reviewer #1: Yes

Reviewer #2: Yes

Reviewer #3: Yes

3. Have the authors made all data underlying the findings in their manuscript fully available?

Reviewer #1: Yes

Reviewer #2: Yes

Reviewer #3: No

4. Is the manuscript presented in an intelligible fashion and written in standard English?

Reviewer #1: Yes

Reviewer #2: Yes

Reviewer #3: Yes

5. Review Comments to the Author

Reviewer #1: General comments

Overall I found this to be a very useful paper for increasing our understanding of the impacts of salinity on American crocodiles in terms of survival, growth, and body condition. The analyses conducted are very informative for Everglades restoration, and I read this study with great interest. I have several general comments to ideally help make the paper easier to follow, and included a list of more details comments below.

In several places, the paper seems geared towards an audience familiar with Florida, but the presumably broader readership would need some more background information. The references are numbered in the text but the reference list is ordered alphabetically. This is an easy fix but made the review a bit more difficult. Also, the introduction would benefit from an expanded discussion on previous research on salinity impacts to crocodiles, and a short summary of the status of crocodiles. The methods could also use some more details to make the analyses that were done easier to understand, and in the results, there were a few sections where I had difficulty in understanding exactly what the models represented (see my more detailed comments below).

Abstract

Line 23, capitalize “we”

What are the units for the body condition? Also is this average for crocodiles, or below or above average?

Line 35, to be clear write “Hypersaline conditions negatively affected”

Introduction

The references are numbered in the text but the reference list is ordered alphabetically, making it difficult to evaluate the references that are used. Please use one system for both the text and reference list. Also there isn’t any information on the status of crocodiles in Florida. This doesn’t need to be extensive, but a few lines on their status, current and previous threats, etc would be useful information to add.

Line 60, add “the southern estuaries”

Lines 62 to 66, what time frame are you referring to here? Current conditions versus predrainage? Or versus 150 years ago?

Line 68, would all readers know what river of grass refers to?

Lines 74, for the readers not familiar with the Everglades, can you explain why is NE Fl Bay important for crocodiles

Lines 77 to 80, although this is a good start, but it would be useful to have more information on the impacts of salinity on crocodiles. There is previous research on this topic, and it would be helpful to have a short review of what has been done before.

Line 87, can you expand on your hypothesis in this line, for example, what do you mean exactly by a disturbance in hydrological conditions? Just a reduction in freshwater flow, or are you also referring to the timing and the resulting salinity conditions?

Methods

Line 140, spell out ENP the first time you use it.

Lines 179 to 181, how many surveys per year were conducted? Also was the survey effort constant across years in terms of number of people, time spent surveying, etc?

Lines 204 to 213, why did you test out three different models versus testing all the factors in a model? Are the independent factors correlated?

Line 217, how did you determine the crocodiles’ age? From the size category? Or time elapsed between captures? I see this is explained in lines 227 to 228, please add this explanation to the growth rate modeling section as well.

Lines 234, are there no other factors that might have affected your recapture probability? For instance, if your survey effort varied across years, that could influence the likelihood of recapture. Also do any environmental factors or habitat types influence the ability to see or capture crocodiles? Or could crocodiles avoid or move out of habitats of high salinity?

Line 239, I have the same question here, why didn’t you look at effects of habitat or even body condition on survival rates?

Results

Line 271, are there no units for the body condition measure? g/cm3?

This would be appropriate for the discussion section but in line 281, why are crocodiles in better condition in the dry season? Is this linked to prey availability?

Body Condition

Lines 294 to 302, it’s interesting here that the # of days <20 psu had a slight negative effect and is nearly significant. Is there any physiological reason why lower salinity could negatively impact crocodiles?

Adding area to the salinity model didn’t improve the model fit by much, and I’m wondering if area is a proxy for salinity values? All the salinity coefficients became non-significant as well once area was added in, and your salinity results suggest that the values differ across areas. Also did you test out adding in season or habitat to the salinity model? Perhaps that would improve your R squared values.

Growth rates

For the growth rate modeling, if the best age model had a cubic term, why wasn’t this tested in the salinity model? And similarly, for the longitude model? Or is age already included in these models and you’re testing first and second order effects of salinity and longitude? In Table 3 it’s not clear to me what values are included in all the models. Also in the methods you state, “We included longitude (easting) (to indicate physical location of capture) and its quadratic effect as additive and multiplicative effects on growth” – but I don’t see any results from multiplicative effects in the table, just additive effects.

Survival analysis

For the recapture rates what does captures stand for in Table 4? Also did the time -dependent recapture rates not end up in the best set of models?

In line 357, “The salinity model alone did not produce significant effects on survival” – this model is actually an age + salinity model right? And do you mean that the whole model was not significant or just the salinity coefficient ?

Line 357-362, these are interesting results but are the differences in survival rates between areas statistically significant? Also are there no differences in sub adult or adult survival rates between areas?

Discussion

Line 377, I’d change this to be more clear “…we assess in this study the body condition and additional population dynamics including growth and survival, while previous studies assessed the relative density”

Line 386, this is confusing since Cape sounds like it also has a score of less than 2, I’d write, “…NEFB, while CAPE crocodiles were in the best body condition.”

Line 408, is the annual survival rate for juveniles or all age categories?

Lines 429 to 432, how is there a tradeoff at NEFB if growth rates are also slower than other areas?

Line 474, mention that Fox Lakes are near to Cabe Sable

Lines 489 to 490, I’d introduce this idea more clearly sooner in the paper that these different metrics are meant to measure different time scales. For instance line 84 to 85, I’d write this out explicitly that body condition – short-term, growth, intermediate, and survival long-term

Figures & Tables

There aren’t figure legends in the file

Table 1

I’d spell out WCA, NPS, and ENP at least once in the table.

Also, it’d be helpful to have a larger map with some of these areas on it – otherwise it’d be hard for readers not familiar with Florida to know what these projects are entailing.

Table 3

Like I mentioned above, I find it difficult to understand which variables were included in these different models

Figure 1

It would be useful to have Taylor Slough in this figure. Also it may be a problem with the resolution in draft pdf but the colors between flam and nefl or between madb and crl look nearly identical in the legend.

Reviewer #2: I believe the authors did a great job introducing the background and objectives of this project, in addition to analyzing the appropriate abiotic and morphological measurements to support their findings. I have minor suggestions in regards to adding further information to the manuscript:

1) In regards to Table 1, is it possible to include a brief timeline of the initial water control projects of the everglades from 150 years ago? It would be good to include a least some of the principal projects that contributed to the degradation of the Everglades. Such information could be useful for researchers utilizing this publication as a reference for future restoration manuscripts or projects.

2) Line 170: how often do the monitoring stations analyze salinity? This can be easily included in this sentence, as well as provides a background on the quantitative number used for finding the average maximum and minimum salinity.

3) Lines 292-293: It doesn't seem to be discussed in result or discussion sections, but is there a suggestion for the males to have lower body condition than females? In the discussion it states male foraging and females staying in one spot due to nesting which would seem to expose them to continuous higher salinity measures and cause lower body condition in females.

Reviewer #3: An interesting study, worth to be published, that shows the role of American crocodiles as ecological indicators. The work also bring new important data on the effect of salinity and other factors on the body condition, survival and growth of American crocodiles.

I have few comments on the manuscript:

Line 48: references are numbered in the text but not in the references list. In the reference list the references are ordered alphabetically and not by the number attributed in the text. Thus I could not know to which reference each number correspond, and evaluate the relevancy of the references.

Line 54: I recommend the authors to check this recent publication linked to their work:

Labarre D., P. Charruau, W.F.J. Parsons, S. Larocque-Desroches, J.A. Gallardo Cruz. 2020. Major hurricanes affect body condition of American crocodile Crocodylus acutus inhabiting Mexican Caribbean islands. Marine Ecology Progress Series 651: 145-162. https://doi.org/10.3354/meps13425

Line 154: In Figure 1 it seems that NEFB was replaced by NEFL. Is it correct? I think it is a typing error in the figure.

Line 179: indicate if captures were conducted at night, daylight or both. Perhaps indicate how crocodiles are detected for non-crocodile experts.

Line 199-201: see reference above of Labarre et al. (2020) on the effect of hurricanes, seasonal fluctuations of environmental conditions and reproductive events on body condition of C. acutus.

Line 202: did you verified that your data respect the assumptions needed to calculate Fulton’s K? If so, you should add that information (perhaps in SI).

Line 243-244: a parenthesis is missing.

Line 259: insert Mean and SD for BBC and FLAM.

Line 266: insert “]” after “21”.

Line 268: a period is missing.

Line 266-268: ok but what caused the 2015 increase?

Line 274: how did you determined that a K<2.0 is a low body condition?

Line 294: I think that “above” and “≥” are repetitive.

Line 372: I detected several errors with the use of parenthesis and brackets in the discussion.

Line 382-389: check Labarre et al. (2020) for American crocodiles in Mexico.

Figures: titles of figures are not provided or I did not found them.

SI figures: titles of figures are not provided or I did not found them.

The link to download data did not worked.

6. PLOS authors have the option to publish the peer review history of their article (what does this mean?). If published, this will include your full peer review and any attached files.

Reviewer #1: **Yes: **Ruscena Wiederholt

Reviewer #2: No

Reviewer #3: No

---

## [Author Response · Author response to Decision Letter 0]

31 Mar 2021

Dear Editors,

Thank you for providing comments to our manuscript toward improving its clarity and reach. I have addressed each comment and detail how I did so in the revised manuscript. I sincerely appreciate the constructive comments that came from the reviewers.

Required changes:

1. Please address the comments of Reviewer 1 and the AE, who both find that the manuscript is written under the assumption that readers have some familiarity with south Florida and the Everglades. PlosOne is an international journal with a diverse readership, so some more general information is required. For example, Figure 1 showing the study site should provide a larger geographic context, and the text-based description should include some more general location and site information. This need not be extensive, a few sentences will do.

A more general description of location and study site has been provided in the introduction as well as in methods. Thank you for that suggestion.

A detailed Species status and distribution has been included in the introduction. 

Figure 1 now includes species distribution map and location of study site in insert map.

2. Please address the comments and concerns raised by Reviewer 1 regarding data management and statistical analyses. Most of these appear to stem from a lack of detail provided in the manuscript, and may only require changes to the text.

Addressed throughout, see detailed changes in Reviewer 1 comments below.

3. Please balance the presentation of data vs. the output from statistical modeling. As presented the Results in main manuscript deal almost exclusively with tables of model outputs, and the actual data displays are relegated to the Supplemental Material. Figures 1-4 in the Supplemental Material should likely be part of the main document. In contrast, Table 1 is less useful as part of the main document and may be better moved to the SI section.

Figure 1-4 of Supplemental Material have now been moved to the main document and labelled as Figures 1-4. 

Conversely, Table 1 has been moved to the SI section as S1 Table.

1.) Please ensure that your manuscript meets PLOS ONE's style requirements, including those for file naming. The PLOS ONE

style templates can be found at

Adhered to throughout ms.

2.) We note that you have stated that you will provide repository information for your data at acceptance. Should your manuscript be accepted for publication, we will hold it until you provide the relevant accession numbers or DOIs necessary to access your data. If you wish to make changes to your Data Availability statement, please describe these changes in your cover letter and we will update your Data Availability statement to reflect the information you provide.

Data is accessible and live now at 

Briggs-Gonzalez, Venetia (2021), American crocodile captures in South Florida, Dryad, Dataset, https://doi.org/10.5061/dryad.nzs7h44q7

3.) Please include a caption for figure 1, 2, 3 and 4.

Figure captures included for Fig 1,2,3,4. 

4.) We note that Figure 1 in your submission contain map images which may be copyrighted. All PLOS content is published under the Creative Commons Attribution License (CC BY 4.0), which means that the manuscript, images, and Supporting Information files will be freely available online, and any third party is permitted to access, download, copy, distribute, and use these materials in any way, even commercially, with proper attribution. For these reasons, we cannot publish previously copyrighted maps or satellite images created using proprietary data, such as Google software (Google Maps, Street View, and Earth). For more information, see our copyright guidelines: http://journals.plos.org/plosone/s/licenses-and-copyright.

Fig 1 Map was generated from ArcGIS layers of our crocodile survey routes and does not infringe on any previously copyrighted images. We similarly created insert map of species distribution map using species locale records and is provided as location context. 

Reviewer #1: General comments

Overall I found this to be a very useful paper for increasing our understanding of the impacts of salinity on American crocodiles in terms of survival, growth, and body condition. The analyses conducted are very informative for Everglades restoration, and I read this study with great interest. I have several general comments to ideally help make the paper easier to follow, and included a list of more details comments below. In several places, the paper seems geared towards an audience familiar with Florida, but the presumably broader readership would need some more background information. The references are numbered in the text but the reference list is ordered alphabetically. This is an easy fix but made the review a bit more difficult. Also, the introduction would benefit from an expanded discussion on previous research on salinity impacts to crocodiles, and a short summary of the status of crocodiles. The methods could also use some more details to make the analyses that were done easier to understand, and in the results, there were a few sections where I had difficulty in understanding exactly what the models represented (see my more detailed comments below).

Abstract

Line 23, capitalize “we” 

Addressed

What are the units for the body condition? 

Fulton’s Body condition factor does not have units – it is a mass/length relationship and a scaling factor by 100 and is described in Body condition portion of methods.

Also is this average for crocodiles, or below or above average? 

Reworded to read Mean body condition in this study throughout ms. This study’s mean value is compared to body condition indices of the same species as well as other crocodilians in other locations in the discussion.

Line 35, to be clear write “Hypersaline conditions negatively affected” 

Addressed.

Introduction

The references are numbered in the text but the reference list is ordered alphabetically, making it difficult to evaluate the references that are used. Please use one system for both the text and reference list. Also there isn’t any information on the status of crocodiles in Florida. This doesn’t need to be extensive, but a few lines on their status, current and previous threats, etc would be useful information to add.

Addressed: in-text citations are numbered and match the reference list. 

The status of the American crocodile is provided in the introduction. 

Line 60, add “the southern estuaries” 

Addressed

Lines 62 to 66, what time frame are you referring to here? Current conditions versus predrainage? Or versus 150 years ago? 

Timeframe is current conditions vs pre-drainage. 

Line 68, would all readers know what river of grass refers to? 

Clarified Everglades as the River of Grass as appropriately coined by Marjorie Stoneman Douglas in 1947.

Lines 74, for the readers not familiar with the Everglades, can you explain why is NE Fl Bay important for crocodiles 

Included importance of NE Fl Bay as nesting population and addressed in the introduction and discussion.

Lines 77 to 80, although this is a good start, but it would be useful to have more information on the impacts of salinity on crocodiles. There is previous research on this topic, and it would be helpful to have a short review of what has been done before.

Salinity impacts on American crocodiles included in introduction and as discussion points.

Line 87, can you expand on your hypothesis in this line, for example, what do you mean exactly by a disturbance in hydrological conditions? Just a reduction in freshwater flow, or are you also referring to the timing and the resulting salinity conditions? 

Clarified in introduction.

Methods

Line 140, spell out ENP the first time you use it.

Addressed

Lines 179 to 181, how many surveys per year were conducted? Also was the survey effort constant across years in terms of number of people, time spent surveying, etc?

Capture surveys detailed in methods.

Lines 204 to 213, why did you test out three different models versus testing all the factors in a model? Are the independent factors correlated? 

We approached it as additive and wanted to parse out effects instead of putting them all together from the beginning. In the end salinity was tied to area but area effects were stronger.

Line 217, how did you determine the crocodiles’ age? From the size category? Or time elapsed between captures? I see this is explained in lines 227 to 228, please add this explanation to the growth rate modeling section as well. 

Growth determined as time elapsed between captures, explanation added to growth rate section.

Lines 234, are there no other factors that might have affected your recapture probability? For instance, if your survey effort varied across years, that could influence the likelihood of recapture. Also do any environmental factors or habitat types influence the ability to see or capture crocodiles? Or could crocodiles avoid or move out of habitats of high salinity? 

Recapture probability was dependent on presence of crocodiles. Consistent effort was made to capture crocodiles. Detectability of crocodiles can depend on air and water temperature and lunar cycle and this is across years but we accounted for this by conducting surveys during the absence of high winds, not on full moon and at low tide which would limit accessibility to shorelines. Crocodiles can move out of habitats of high salinity within an area or between recapture events (but not likely between areas) hence why capture location was used as a snapshot of time.

Line 239, I have the same question here, why didn’t you look at effects of habitat or even body condition on survival rates? 

Crocodiles move between habitat types within an area over their lifetime and the points of capture and recaptures provide snapshots into where crocodiles were at that time but does not necessarily indicate where they spend all of their lifetime. Good idea on evaluating the effects of body condition on survival rates – we intend to next investigate the relationships between crocodile population parameters and how they affect crocodiles in South Florida.

Results

Line 271, are there no units for the body condition measure? g/cm3? Fulton’s condition factor does not carry units. 

It is a value typically between 0 and 4 and is scaled by multiplying by 100.

This would be appropriate for the discussion section but in line 281, why are crocodiles in better condition in the dry season? Is this linked to prey availability? 

Dry season concentrates prey, wet season prey is dispersed. Everglades crocodiles experience seasonal fluctuations of feast and famine, the nature of this dynamic is explained in the discussion.

Body Condition

Lines 294 to 302, it’s interesting here that the # of days <20 psu had a slight negative effect and is nearly significant. Is there any physiological reason why lower salinity could negatively impact crocodiles? 

Salinity can also be an indicator of rainfall and may relate to a longer hydroperiod when prey are dispersed affecting feeding ability and ultimately body condition. It will be good to incorporate more factors and their relationships in the future.

Adding area to the salinity model didn’t improve the model fit by much, and I’m wondering if area is a proxy for salinity values? All the salinity coefficients became non-significant as well once area was added in, and your salinity results suggest that the values differ across areas. Also did you test out adding in season or habitat to the salinity model? Perhaps that would improve your R squared values. 

Area is a good proxy for salinity and crocodiles differ between areas, and within an area salinity may have further effects on a finer scale. Habitat and seasonal effects were tested in the basic model and we focused on effects of salinity and area effects in further models.

Growth rates

For the growth rate modeling, if the best age model had a cubic term, why wasn’t this tested in the salinity model? And similarly, for the longitude model? Or is age already included in these models and you’re testing first and second order effects of salinity and longitude? 

Growth is a direct relationship to Age and is included in all models and focus on investigating first and second order effects of salinity and longitude (i.e., area/location of capture) on crocodile growth.

In Table 3 it’s not clear to me what values are included in all the models. 

Growth curve analysis was conducted on growth increments between initial capture at hatchling size and first and subsequent recapture. 

No other values were used to construct the crocodile growth curve. By adding sex, growth curves differentiated between males and females; when salinity was added growth was affected and curves differed under hypersaline conditions within the first year, after 5 years and after 10 years of exposure to hypersaline conditions. Similarly when longitude (location of crocodile capture) was added to growth analyses, growth differed in crocs among the locations of capture with crocs from NEFB having the slowest estimated growth vs crocs from Cape having the most improved growth. 

Also in the methods you state, “We included longitude (easting) (to indicate physical location of capture) and its quadratic effect as additive and multiplicative effects on growth” – but I don’t see any results from multiplicative effects in the table, just additive effects.

This should be additive effects of longitude only – edited in methods.

Survival analysis

For the recapture rates what does captures stand for in Table 4? Recapture rate is the probability of capturing a crocodile within an age class. Also did the time -dependent recapture rates not end up in the best set of models? 

Age is a more reliable indicator of survival relative to time and time did not comprise the best models for survival analysis.

In line 357, “The salinity model alone did not produce significant effects on survival” – this model is actually an age + salinity model right? And do you mean that the whole model was not significant or just the salinity coefficient? 

Both salinity alone and salinity and area were tested but area effects when tested alone diluted any salinity effects including when in combination. 

Line 357-362, these are interesting results but are the differences in survival rates between areas statistically significant? Also are there no differences in sub adult or adult survival rates between areas? Survival rates differed between areas (p<0.05). Overall survival rates for subadults neared 90% and for adults was 90% plus and we did not assess differences in survival rates between age classes by areas for this paper.

Discussion

Line 377, I’d change this to be more clear “…we assess in this study the body condition and additional population dynamics including growth and survival, while previous studies assessed the relative density”

Addressed

Line 386, this is confusing since Cape sounds like it also has a score of less than 2, I’d write, “…NEFB, while CAPE crocodiles were in the best body condition.”

Addressed

Line 408, is the annual survival rate for juveniles or all age categories?

Here, survival rate refers to hatchling survival to one year while becoming a juvenile.

Lines 429 to 432, how is there a tradeoff at NEFB if growth rates are also slower than other areas?

Clarified that NEFB was disadvantaged for all measures.

Line 474, mention that Fox Lakes are near to Cabe Sable

Addressed

Lines 489 to 490, I’d introduce this idea more clearly sooner in the paper that these different metrics are meant to measure different time scales. For instance line 84 to 85, I’d write this out explicitly that body condition – short-term, growth, intermediate, and survival long-term

Introduced earlier in introduction.

Figures & Tables

There aren’t figure legends in the file

Figure legends provided for all figures throughout revised ms.

Table 1

I’d spell out WCA, NPS, and ENP at least once in the table.

Addressed in new S1 Table.

Also, it’d be helpful to have a larger map with some of these areas on it – otherwise it’d be hard for readers not familiar with Florida to know what these projects are entailing.

Larger distribution map included and some relevant points labelled in Figure 1

Table 3

Like I mentioned above, I find it difficult to understand which variables were included in these different models

Growth is calculated as size at hatching subtracted from size at recapture. The models then further looked at growth between males and females in the “sex’ model, growth under salinity conditions in the “salinity” model and growth of crocs between areas in the “longitude” model.

Figure 1

It would be useful to have Taylor Slough in this figure. Also it may be a problem with the resolution in draft pdf but the colors between flam and nefl or between madb and crl look nearly identical in the legend.

Figure 1 has Taylor Slough included and the colors of survey routes by area have been revised to show more clearly.

Reviewer #2: I believe the authors did a great job introducing the background and objectives of this project, in addition to analyzing the appropriate abiotic and morphological measurements to support their findings. I have minor suggestions in regards to adding further information to the manuscript:

1) In regards to Table 1, is it possible to include a brief timeline of the initial water control projects of the everglades from 150 years ago? It would be good to include a least some of the principal projects that contributed to the degradation of the Everglades. Such information could be useful for researchers utilizing this publication as a reference for future restoration manuscripts or projects. 

This would indeed be good to have but there have been numerous water control projects in the Everglades 150 years and beyond and in itself is a manuscript. We focus on the restoration efforts of current times and focus on crocodilian responses.

2) Line 170: how often do the monitoring stations analyze salinity? This can be easily included in this sentence, as well as provides a background on the quantitative number used for finding the average maximum and minimum salinity.

Salinity is recorded hourly, we calculated mean daily salinity from these readings, included in methods.

3) Lines 292-293: It doesn't seem to be discussed in result or discussion sections, but is there a suggestion for the males to have lower body condition than females? In the discussion it states male foraging and females staying in one spot due to nesting which would seem to expose them to continuous higher salinity measures and cause lower body condition in females.

Males had poorer body condition scores than females, most likely because females are preparing for nesting and have invested fat reserves toward egg production and cannot “afford” to be in poor condition for successful nesting to occur. Nesting females have been observed at higher salinity areas as a tradeoff between nesting site access and high salinity conditions. Males do not undergo the same physiological energy expense of egg production and nesting and poor body condition may be more apparent. Discussed in results and discussion.

Reviewer #3: An interesting study, worth to be published, that shows the role of American crocodiles as ecological indicators. The work also bring new important data on the effect of salinity and other factors on the body condition, survival and growth of American crocodiles.

I have few comments o n the manuscript:

Line 48: references are numbered in the t ext but not in the references list. In the reference list the references are ordered alphabetically and not by the number attributed in the text. Thus I could not know to which reference each number correspond, and evaluate the relevancy of the references.

Sincerest apologies, references have been numbered in both text and references list

Line 54: I recommend the authors to check this recent pu blication linked to their work:

Labarre D., P. Charruau, W.F.J. Parsons, S. Larocque-Desroches, J.A. Gallardo Cruz. 202 0. Major hurricanes affect body condition of American crocodile Crocodylus acutus inhabiting Mexican Caribbean islands. Marine Ecology Progress Series 651: 145-162. https://doi.org/10.3354/meps13425

Excellent inclusion to ms in introduction and discussion. Thank you.

Line 154: In Figure 1 it seems that NEFB was replaced by NEFL. Is it correct? I think it is a typing error in the figure. 

All NEFL have been replaced by NEFB throughout the manuscript in text and on figures

Line 179: indicate if captures were conducted at night, daylight or both. Perhaps indicate how crocodiles are detected for noncrocodile experts.

Nocturnal surveys and crocodiles detected using eyeshine explained in methods.

Line 199-201: see reference above of Labarre et al. (2020) on the effect of hurricanes, seasonal fluctuations of environmental conditions and reproductive events on body condition of C. acutus. 

Included in introduction.

Line 202: did you verified that your data respect the assumptions needed to calculate Fulton’s K? If so, you should add that information (perhaps in SI). Yes, data met assumptions of normality to calculate Fulton’s K and to be used in multivariate regression analysis. Data were similarly used in previous studies referenced and explained in methods and results. 

Line 243-244: a parenthesis is missing. Addressed

Line 259: insert Mean and SD for BBC and FLAM. Addressed and included for all areas.

Line 266: insert “]” after “21”. Addressed.

Line 268: a period is missing. Addressed.

Line 266-268: ok but what ca used the 2015 increase? 

Discussed that 3-4 years post project initiation had some reduced effects of salinity in discussion.

Line 274: how did you determined that a K<2.0 is a low body condition? 

We calculated reference quartiles in a separate study and determined ideal body condition for crocodiles in south Florida to be >2.4, acceptable at a K≥2.0 and poor K< 2.0 and presented in both results and disucssion.

Line 294: I think that “above” and “≥” are repetitive. 

Omitted “above”.

Line 372: I detected several errors with the use of pa renthesis and brackets in the discussion. 

Parentheses were used for in text citations and were converted to numbered citations using brackets. This matter has been addressed throughout ms as numbered citations were re-ordered and literature cited presented in numbered form.

Line 382-389: check Labarre et al. (2020) for American crocodiles in Mexico.

Included in introduction and discussion.

Figures: titles of figures are not provided or I did not found them.

Figure titles included throughout manuscript.

SI figures: titles of figures are not provided or I did not found them .

SI Figure titles provided in mansucript

The link to download data did not worked. 

That was a preliminary data link – Data available at Dryad Digital Repository https://doi.org/10.5061/dryad.nzs7h44q7 (Briggs-Gonzalez et al., 2020).

---

## [Decision Letter · Decision Letter 1]

8 Apr 2021

American crocodiles (Crocodylus acutus) as restoration bioindicators in the Florida Everglades

PONE-D-20-34204R1

Dear Dr. Briggs-Gonzalez,

We’re pleased to inform you that your manuscript has been judged scientifically suitable for publication and will be formally accepted for publication once it meets all outstanding technical requirements.

Kind regards,

Christopher M. Somers

Academic Editor

PLOS ONE

Additional Editor Comments (optional):

Reviewers' comments:

Reviewer's Responses to Questions

**Comments to the Author**

1. If the authors have adequately addressed your comments raised in a previous round of review and you feel that this manuscript is now acceptable for publication, you may indicate that here to bypass the “Comments to the Author” section, enter your conflict of interest statement in the “Confidential to Editor” section, and submit your "Accept" recommendation.

Reviewer #1: All comments have been addressed

2. Is the manuscript technically sound, and do the data support the conclusions?

Reviewer #1: Yes

3. Has the statistical analysis been performed appropriately and rigorously? 

Reviewer #1: Yes

4. Have the authors made all data underlying the findings in their manuscript fully available?

Reviewer #1: Yes

5. Is the manuscript presented in an intelligible fashion and written in standard English?

Reviewer #1: Yes

6. Review Comments to the Author

Reviewer #1: The authors have addressed the majority of my comments/concerns. I did still have some concerns about the testing factors individually in statistical tests vs combining them, but I seem to be the sole reviewer with this concern.

7. PLOS authors have the option to publish the peer review history of their article (what does this mean?). If published, this will include your full peer review and any attached files.

Reviewer #1: No

---

## [Editor Report · Acceptance letter]

21 Apr 2021

PONE-D-20-34204R1 

American crocodiles (*Crocodylus acutus*) as restoration bioindicators in the Florida Everglades 

Dear Dr. Briggs-Gonzalez:

I'm pleased to inform you that your manuscript has been deemed suitable for publication in PLOS ONE. Congratulations! Your manuscript is now with our production department. 

Kind regards, 

on behalf of

Dr. Christopher M. Somers 

Academic Editor

PLOS ONE